# Commuting in Urban Kenya: Unpacking Travel Demand in Large and Small Kenyan Cities

**Deborah Salon** [1,*]  **and Sumila Gulyani** [2]

[1]    School of Geographical Sciences and Urban Planning, Arizona State University, Tempe, AZ 85281, USA
[2]    World Bank, Washington, DC 20433, USA
*    Correspondence: deborah.salon@asu.edu

**Abstract:** In Kenya's capital city, Nairobi, streets are regularly gridlocked. While it is clear that roads are congested at peak hours, it is not known which commuters are experiencing that congestion or what their commute times actually are. Even less is known about commuting patterns in other Kenyan cities. This paper contributes new evidence on commuting from a survey of 14,580 households, conducted in 15 Kenyan cities in 2013. Walking and matatus—privately-operated paratransit—account for 89% of all adult commuting in urban Kenya. As cities increase in size, the proportion relying on walking falls and matatu use increases. Within a city, commuters with higher income and education, and those living further from the city center, are more likely to use matatus rather than walk. Commute times are surprisingly short. In smaller Kenyan cities the median commute time is just 20 min. In Nairobi, the median commute time is 30 min, and only 5% of those surveyed reported commuting an hour or longer. These data paint a remarkably sustainable picture of urban travel patterns in Kenya. As incomes, education levels, and demand for motorized travel rise, the challenge will be to expand and improve the system while maintaining its sustainability.

**Keywords:** transport planning; commute time; commute mode; poverty; matatu; Nairobi

## 1. Introduction

News reports as well as some studies suggest that commuting in Kenya's capital city of Nairobi is a nightmare. Streets are gridlocked with a mix of cars, trucks, and matatus—the privately-owned and operated paratransit vehicles that serve as the city's primary means of public transport. Reports claim that commutes are regularly more than 2 h each way [1], and the combined economic cost of this wasted time to the city is substantial [2–4].

There are also reports of the large percentage of commuters who are not stuck in traffic because the only means of transport they can afford is their own feet. These commuters are walking, often on footpaths alongside the gridlocked traffic, and therefore often also subject to high levels of air pollutants [5] as well as the inherent danger of being pedestrians in a tense traffic environment. Kenya has one of the highest pedestrian fatality rates in the world [6].

A large fraction of the vehicles plying the roadways in Nairobi are matatus. These privately owned and operated passenger vans and minibuses provide public transport service for approximately 1 million people each day. Matatus are infamous for sometimes driving dangerously as they jockey for passengers that wait by the side of the road, and there are also complaints about unpredictable fares (operators often raise the price in times of high demand) [7,8]. But matatus are ubiquitous, and they provide high-frequency [9] transport service without government subsidy that is affordable for many (but not all) residents.

Variants of this story are common in major cities throughout the Global South [10–12]. Despite obvious problems, an enviable feature of these urban transport systems is their small environmental

footprint. Walking is the cheapest and most environmentally friendly form of transport, and it improves physical fitness as well [13]. When their seats are filled, buses and vans are highly energy efficient per passenger kilometer of travel [14]. Ubiquitous public transport vehicles ensure short walk access and wait times. The challenge in these cities, then, is to maintain environmental sustainability while solving the safety, health, and commuting time problems caused by their current, relatively informal, systems. To create a package of policies and investments that can meet this challenge, it is important to know how people are traveling today and who is experiencing each of the problems outlined above.

For the case of urban Kenya, this raises the following questions. What exactly is the share of walking and matatu use in Nairobi? How different is the modal split in smaller cities in Kenya? What factors influence modal choice, especially the decision to opt for environmentally friendly modes such as walking, biking or public transport? This paper aims to address these questions and fill some of the important gaps in our understanding of travel demand in urban Kenya. It does so by presenting analysis of data from a rare and in-depth survey of 14,580 urban households in 15 cities, conducted in 2013 under the aegis of the World Bank. It adds to the existing literature on travel in Nairobi, and contributes the first analysis of commuting in 14 additional Kenyan cities.

We document who is commuting and who is not, the modes commuters use, and how much time and money they spend. At the household level, we look at access to motorized transport (mainly matatus). At the individual level, we look at differences between men and women, and across wealth, age, and education levels.

The descriptive data are important in themselves because they provide a first-of-its-kind overview of commuting in urban Kenya. These data reveal that commuting in urban Kenya is dominated by walking and matatus. Surprisingly, matatu services are widely available not just in Nairobi but also across urban Kenya—across 15 Kenyan cities, 92% of households report having matatu service available near their homes, and this rises to 98% in Nairobi. Among adult commuters, 42% walk, 48% ride a matatu, 5% commute by private car, and the remaining 5% use other modes (company transport, two-wheelers, etc.). Despite the ubiquity of matatu services, in 12 of the 15 cities surveyed, the majority walk. In the two largest cities—Nairobi and Mombasa—the majority commute by matatu. In Eldoret, commuting is evenly split between walk and matatu. Contrary to previous research and popular perception, we find that reported commute times are relatively short. Overall, then, the current travel choices of urban Kenyans appear to be rather environmentally friendly.

We use multivariate analyses to shed light on the key factors that influence travel demand—that is, whether urban Kenyans commute and the modes they use—and discuss how these differ between cities. This analysis suggests that income, education level, and distance from the city center are associated with a higher likelihood of matatu commuting. This means that as the Kenyan economy grows and cities increase in size, the demand for motorized transport will also increase. If matatu service cannot keep up with demand, increasing numbers of urban Kenyans may turn to private vehicles (cars and motorcycles), reducing the overall sustainability of the transport system.

To expand and improve public transport service along with growth in demand, some experts have suggested that the government step into the matatu market with larger buses and dedicated infrastructure [15]. Our work provides data that will help Kenyan officials that are considering this option. We use the 2013 survey data to estimate the size of the matatu market in each city, providing rough estimates for two key pieces of information for public transport planners: The total number of commute rides served by matatus each day, and the associated fare revenue.

This paper continues by reviewing prior evidence on the experience of commuting in urban Africa generally and in Nairobi, Kenya specifically. Then, it proceeds to present findings on motorized transport access, commute modes, commute times, and the matatu market in urban Kenya from the 2013 World Bank survey data. We conclude with an integrative discussion of these findings, what they suggest for policy, and directions for future research.

## 2. Prior Evidence on Commuting in Urban Africa

High quality data on commuting in urban Africa is scarce. Large-scale travel surveys are rarely collected, so many of the insights we have are based on smaller surveys that may not be fully representative of the relevant population. Nearly all prior surveys were collected in capital or primary cities, meaning that there is virtually no data on commuting in the many smaller cities in Africa.

From these existing data, four key findings emerge. First, a large fraction of commuters in urban Africa travel on foot. In Nairobi, estimates suggest that approximately one-third of adult commuters walk [16–19]. In Addis-Ababa, Ethiopia, Lusaka, Zambia, and Kampala, Uganda, this figure is closer to 70% [20,21]. This compares to a figure of 60% walk commute mode share in Dakar, Senegal in 2001, and 24% in Capetown, South Africa in 2001 [20]. In 2013, 21% of all workers in South Africa walked to work [22].

Second, traffic congestion is substantial, and even motorized modes do not move swiftly through African urban centers at peak hours. In Nairobi, traffic models estimated vehicle speeds to be 8.3 km/hr during the morning peak and 7.6 km/hr during the evening peak [23]. Daytime average roadway speeds in Dar es Salaam, Tanzania were 8–15 km/hr [24].

As Gonzales et al. [23] explain, congestion occurs in these cities because the roadway network is simply much smaller than you would find in cities of similar populations in wealthier places. Nairobi's road network has the additional problem of a lack of alternate routes; if there is a crash or other blockage on one major road, often everyone is simply stuck. The lack in total road capacity together with a lack of alternate routes causes serious congestion even at low car ownership levels.

Third, commute times are long and/or highly unpredictable due to congestion patterns. Andreasen and Møller-Jensen [25] write, "A trip to the centre [in Dar es Salaam, Tanzania] . . . can take anything from 30 min to 3 h depending . . . on the traffic conditions." (pp. 23–24).

Finally, with a focus on slum residents in Nairobi, Salon and Gulyani found that gender played a large role in determining both who commuted and whether they used motorized modes [26]. Women were less likely to use motorized modes, even after controlling for differences between men and women in both childcare responsibilities and education levels.

## 3. Data

The data that are the basis for this paper were collected under contract to the World Bank in 2013. Enumerators surveyed a random sample of approximately 1000 Kenyan households in each of 15 cities (see Figure 1), for a total of 14,580 completed surveys. To obtain a random sample, the team first worked with the Kenya National Bureau of Statistics to select enumeration areas (the smallest geographic census unit in Kenya, similar to census blocks in the United States) in each city. Then, enumerators developed a sampling frame for each selected enumeration area by walking the neighborhoods and creating an updated list of all households. Finally, enumerators randomly selected households to interview in each enumeration area from this sampling frame. The surveys were conducted in person at the household's home, and questions covered a broad range of topics related to household income, expenditure and assets, housing, infrastructure access, and age, education, and commuting information for all household members. Full details about the survey methodology can be found in the World Bank survey overview report [27].

The following four key questions about transportation access and commuting were included on the survey:

1.　What is your main mode of travel for work/school? (asked of all household members over 4 years old who work or study)
2.　How many minutes does it take to get to work or school? (asked of all household members over 4 years old who commute)
3.　Is there a transportation service, such as bus or matatu, into the city center in your neighborhood or within a 20-minute walk?

4.　Does your household currently own any ... Private Cars? Motorcycles? Bicycles?

For the analyses in this article, we combine respondent answers to these questions with household sociodemographic and home location characteristics to paint a picture of commuting in urban Kenya.

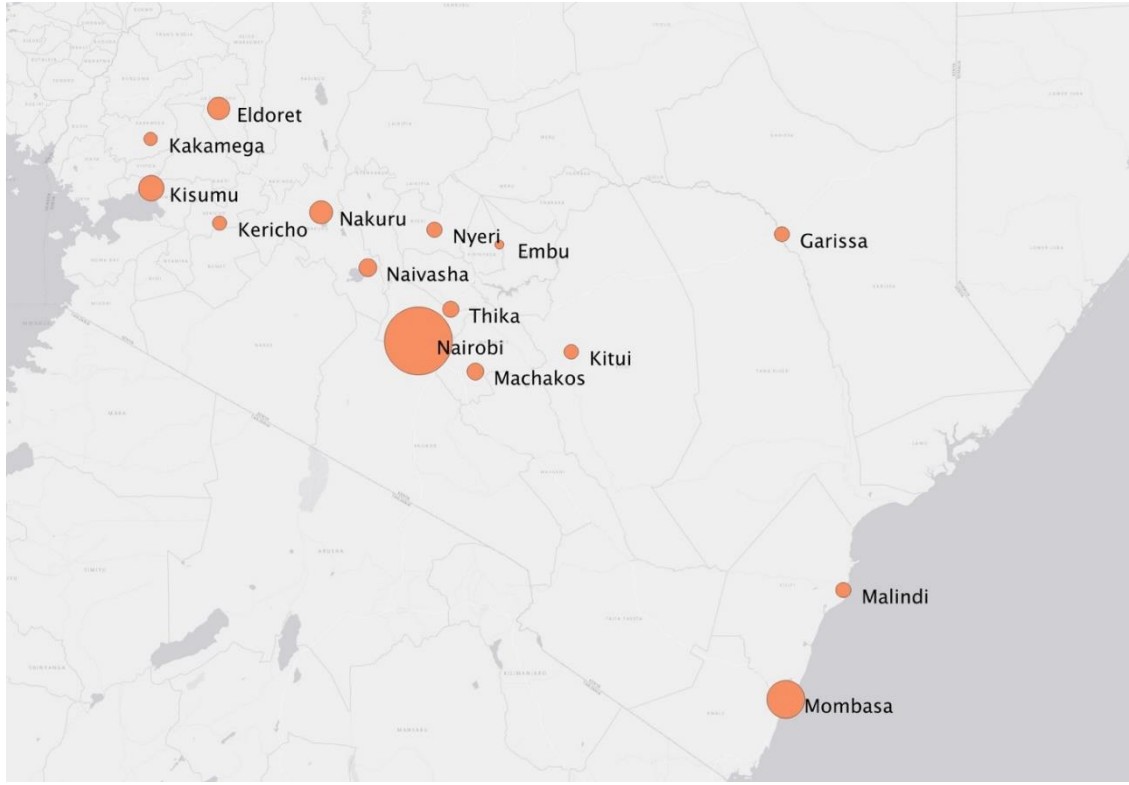

**Figure 1.** Locations of 15 surveyed cities in Kenya. Dot size proportional to population.

*3.1. Representativeness of Data: Comparison with Other Data Sets on Commuting in Nairobi, Kenya*

To our knowledge, there have been five surveys conducted in Nairobi since 2004 that include transport-related questions and aim to collect data from a representative sample of the city's population. These include two surveys by the Japan International Cooperation Agency (JICA) in 2004 and 2013, one survey by the Kenya Institute for Public Policy Research and Analysis (KIPPRA) in 2004, a 2010 survey by the African Centre of Excellence for Studies in Public and Non-Motorised Transport (ACET), and the survey conducted by the World Bank in 2013 that is the basis of this research. In addition, the 2009 Kenya Census reports the percent of households that own a "car/truck/tuk tuk" (a tuk tuk is a 3-wheeled motorized vehicle). We are aware of no surveys other than the 2013 World Bank effort that collected commuting data in Kenya's other cities and towns.

A quick look at Table 1 reveals some striking differences between the Nairobi results of these data collection efforts. Comparing the 2004 KIPPRA and JICA surveys, there are substantial differences in both vehicle ownership levels and the percent of commuters who walk to work. A comparison between the 2013 JICA survey, the 2010 ACET survey, and the 2013 World Bank survey yields similar discrepancies. In both cases, the JICA sample includes more car owners and fewer walk commuters. In addition, the 2013 JICA survey average travel times are substantially longer than those reported in the 2013 World Bank survey, even when disaggregated by mode. In spite of these differences, both of these datasets indicate that average travel times in Nairobi are similar for different motorized modes (cars and matatus), and that the average walk trip is approximately half the duration of the average motorized trip.

**Table 1.** A comparison of the evidence on commuting in Nairobi, Kenya.

| Survey | Motorized Access [1] | Vehicle Ownership | Commute on Foot (Adults) | Commute Time (Adults) |
|---|---|---|---|---|
| 2004 KIPPRA [18] | 80% | 14% of HH | 38% | NA |
| 2004 JICA [16,28] | NA | 21% of HH | 29% | NA |
| 2010 ACET [29] | NA | 13% of HH | 41% (includes bicycle) | NA |
| 2013 JICA [17] | 74% | 25% of HH | 28% | 49 min (avg) 40 min (median) |
| 2013 World Bank [19] | 64% | 8% of HH | 36% | 31 min (avg) 30 min (median) |
| 2009 Kenya Census [30] | NA | 12% of HH | NA | NA |

[1] Households with at least one member who uses a motorized mode. NA = Not Available to this research team.

Understanding these differences is important, especially since our analysis of the 2013 World Bank survey leads to conclusions about commuting in Nairobi that are somewhat different from the conventional wisdom. First, we considered the sampling plans for each survey. The researchers who administered both the 2004 KIPPRA survey and the 2013 World Bank survey aimed to obtain a representative sample of Nairobi households, and each provides documentation of these efforts. The JICA survey documentation indicates that their samples were designed to be geographically representative of the population, but how samples were selected within each geographical area is not explained. The limited documentation available on the ACET survey also states that they worked to obtain a representative sample of Nairobi's population, but the methods used are not described in the available documentation.

To investigate further, we turned to the 2009 Kenya Census. The Census had only one question that directly compares to the samples here—whether a household owned a "car/truck/tuk tuk". In 2009, the Census reported that 12% of households in Nairobi owned one of these. If we can trust the Census, then, it appears that on the metric of car ownership, the 2013 World Bank survey sample is biased against car-owning households, and both JICA samples as well as the KIPPRA sample are biased toward car-owning households. The ACET sample is consistent with the Kenya Census.

The 2009 Census also included a number of questions unrelated to transport that were also asked on the 2013 World Bank survey. These were materials of the home structure and access to water, electric lighting, and sanitation. On these metrics, the 2013 World Bank survey responses matched the 2009 Census well—the 2013 data indicated a slight improvement over the 2009 data for Nairobi. Based on this comparison with the Census, the 2013 World Bank sample appears to be representative of the population of Nairobi on most metrics, but to have undersampled car-owning households.

*3.2. Profile of Households in Kenyan Cities*

Table 2 summarizes key indicators for the five largest Kenyan cities—Nairobi, Mombasa, Nakuru, Kisumu, and Eldoret—and for the rest of urban Kenya (RUK). All survey-based indicators were calculated using household weights to correct for stratification in the sampling method. These data show both similarities and differences across cities in Kenya. Urban Kenyan households are poor, with the poverty rate hovering around the 50% mark for most cities. Household sizes are not large at approximately 3 people per household. Most urban Kenyan households rent their homes.

Nairobi is the nation's capital, densely populated and by far the largest city, located in the central highlands of the country. Nairobi's residents pay high rents, and compared to other Kenyan cities, the population is relatively more educated and household incomes are higher. A large percentage of the population (41%) resides in informal settlements (also known as slums).

Mombasa is a port city, located on the coast and has a culture that is heavily influenced by Arab immigrants. Mombasa is a mid-sized, densely populated city. Residents are poorer than those in Nairobi, but a smaller fraction of Mombasa residents live in informal settlements (26%). The percent of

women in the work force is low (see Table 4), and the percent of highly-educated households is only 20%—surprisingly low compared to Nairobi's 37% (see Table 2).

**Table 2.** City indicators and profile of households surveyed.

| | Nairobi | Mombasa | Nakuru | Kisumu | Eldoret | RUK [1] |
|---|---|---|---|---|---|---|
| Population (core city only) [31] | 3,133,518 | 915,101 | 286,411 | 259,258 | 252,061 | n/a |
| Population density (persons/km$^2$, urban area) [31] | 4515 | 4292 | 971 | 1181 | 1344 | n/a |
| Households with car/truck/tuk tuk [30] | 12% | 10% | 8% | 7% | 10% | n/a |
| Total households surveyed | 1182 | 1095 | 1095 | 740 | 976 | 9495 |
| Percent below poverty [2] | 33% | 45% | 52% | 49% | 58% | 51% |
| Percent of households in slums | 41% | 26% | 5% | 60% | 30% | 8% |
| Households with children under 15 | 55% | 51% | 56% | 68% | 53% | 55% |
| Average household size | 3.1 | 2.8 | 3.0 | 3.8 | 3.0 | 3.0 |
| Households where at least one member completed high school | 72% | 63% | 61% | 55% | 63% | 58% |
| Households where at least one member completed college | 37% | 20% | 26% | 25% | 26% | 24% |
| Percent renter households | 90% | 84% | 87% | 70% | 83% | 69% |
| Median home rent (Ksh) | 2500 | 2000 | 1500 | 1700 | 1500 | 1700 |
| Average rent as percent of income | 29% | 20% | 19% | 22% | 24% | 20% |

[1] RUK = Rest of Urban Kenya (the remaining 10 cities in this sample). [2] This indicates whether the household's reported monthly income is greater or less than the Kenyan urban poverty line, calculated for each household based on the number of adults, children 5–14, and children under 5 in the household. The poverty line is not adjusted for differences in the cost of living between cities.

Nakuru and Eldoret are relatively smaller cities in the Rift Valley of western Kenya. Kisumu is an inland port city located on the shore of Lake Victoria in western Kenya. They are poorer and much less densely populated than Nairobi and Mombasa. Kisumu households tend to have more children than households in other Kenyan cities.

The remaining 10 cities (henceforth often referred to as Rest of Urban Kenya or RUK) included in the 2013 survey are smaller in population, but similar to Nakuru, Eldoret, and Kisumu in terms of education and employment levels, as well as poverty levels. The biggest differences relate to housing quality and affordability.

The remainder of this paper summarizes key bivariate and multivariate relationships in these data relating to motorized transport access, commute mode choice, and commute times, highlighting policy-relevant insights that they provide.

## 4. Household Access to Motorized Transport is Highest in the Largest Cities

To understand commuting patterns in urban Kenya, we start with the household as the unit of analysis. Households reported whether they owned motorized vehicles and bicycles. Households also reported whether public transport was available within a 20 minute walk from their house and whether at least one member in their household used it.

We find that although only a minority of households own cars or bicycles, physical availability of public transport is extremely high in urban Kenya. More than 90% of households reported availability across all cities, and in Nairobi, 98% of respondents reported availability of public transport in their vicinity (see Table 3).

As in the case of many public transport systems worldwide, however, physical availability does not necessarily equate to use. Only 60% of households in Nairobi and Mombasa have a member (adult or child) who commutes using a matatu (see Table 3). This statistic ranges from 40–51% in the three mid-sized Kenyan cities, and drops to just 21% in the rest of urban Kenya. Using a framework developed by Gulyani et al. [32], the proportion of households actually using matatu transport can be seen to have "effective access" to public transport, while those reporting availability of public transport in their vicinity can be deemed to have "nominal access."

**Table 3.** Household access to transport.

|  | Nairobi | Mombasa | Nakuru | Kisumu | Eldoret | RUK |
|---|---|---|---|---|---|---|
| Matatu or bus transport to city center available within 20 minute walk? (a) | 98% | 90% | 83% | 86% | 91% | 83% |
| Households with bicycles | 12% | 5% | 14% | 24% | 8% | 8% |
| Households with cars | 8% | 3% | 3% | 4% | 1% | 2% |
| Households where at least one member commutes using a motorized mode | 64% | 61% | 42% | 45% | 52% | 26% |
| Households where at least one member commutes by matatu or bus (b) | 60% | 60% | 40% | 43% | 51% | 21% |
| Gap between availability and use of public transport (a–b) | 38% | 30% | 43% | 43% | 40% | 62% |

The low use of matatu transport relative to availability—that is, the gap between nominal and effective access—is more striking in smaller cities (RUK). This gap could be explained by affordability challenges for many households, but it could also reflect low service frequency or lack of destination coverage. These are common difficulties faced by public transport systems worldwide. In Section 6, we examine some of the factors influencing use of matatus in urban Kenya.

If we broaden our analysis to include all motorized modes and not just matatus, we find that the proportion of households where at least one member commutes using a motorized mode increases by only 1–5 percentage points depending on the city (Table 3). This reflects the small fraction of the population that owns cars or motorcycles and suggests that matatus are, for most urban households, the only means of motorized transport. To gain a more nuanced understanding of household access to motorized transport, we categorized households with at least one commuter by their adult commute mode mix. The categories were:

- Walk-Only: Households in which all commuting adults walk
- Mixed-Modes: Households in which one or more adults walk-commute, but at least one adult uses a motorized mode (i.e., car, motorcycle, or matatu)
- Motorized-Only: Households in which all commuting adults use a motorized mode

Figure 2 provides a visual snapshot of the results. Walk-Only households account for approximately one-third of commuter households in the two largest Kenyan cities, just under half of commuter households in three mid-sized cities, and nearly two-thirds of commuter households in the rest of urban Kenya.

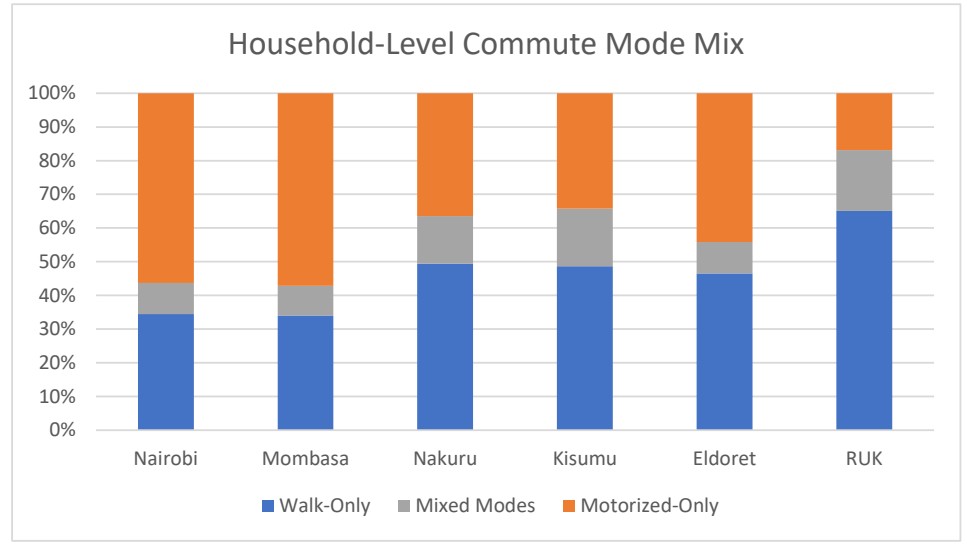

**Figure 2.** Household commute mode split for five Kenyan cities and the rest of urban Kenya.

Figure 3 illustrates the relationship between household-level commute mode mix and poverty. From this figure, it is clear that Walk-Only households are substantially more impoverished than Motorized-Only households everywhere in the country, and Mixed-Mode households are only slightly better off than Walk-Only households in most Kenyan cities. This bivariate analysis suggests that household income, and by extension affordability, might be a key factor influencing mode choice; we test this in Section 6 with multivariate models of commuter status and commute mode choice for adults in urban Kenya.

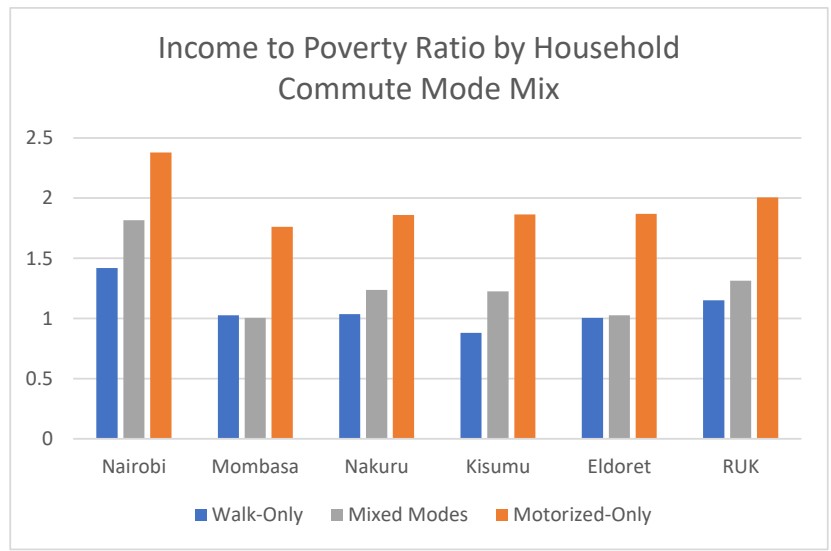

**Figure 3.** Income as a multiple of the household-specific poverty line by commute mode mix in five Kenyan cities and the rest of urban Kenya.

## 5. Adult Commuters Overwhelmingly Walk or Commute by Matatu

In this section, we switch from households as the unit of analysis to focus on individual commuters. Specifically, we analyze our sample of 26,204 working-age adults (i.e., 18–60 years of age), of which 17,518 are commuters, to better understand how they commute. The analysis documents the relative usage of the following modes: Cars, matatus, boda bodas (motorcycle or bicycle taxis), company transport, bicycles, and walking. Because walking is such an important mode, the time spent walking varies substantially, and walk commute times are an important indicator of quality of life, we disaggregate walk commutes into three categories: A short walk of less than 10 min, a medium-length walk of 10–20 min, and a long walk that is more than 20 min. Table 4 presents key indicators for the five biggest cities and rest of urban Kenya.

**Table 4.** Adult commuters: Demographic profile and commuting indicators.

| | Nairobi | Mombasa | Nakuru | Kisumu | Eldoret | RUK [1] |
|---|---|---|---|---|---|---|
| Total adults (18–60) surveyed (N = 26,204) | 2283 | 1954 | 1904 | 1490 | 1747 | 16826 |
| Percent of adults who are students | 12% | 5% | 9% | 13% | 11% | 7% |
| Percent of adults who are employed | 62% | 58% | 61% | 57% | 62% | 59% |
| Percent of adult men who are employed | 76% | 85% | 80% | 73% | 79% | 80% |
| Percent of adult women who are employed | 48% | 31% | 44% | 41% | 44% | 40% |
| Percent of adults commuting for work or study | 73% | 64% | 70% | 69% | 75% | 65% |
| Percent adult commuters using matatus | 53% | 60% | 39% | 36% | 48% | 18% |
| Percent motorized adult commuters | 62% | 63% | 41% | 40% | 51% | 25% |
| Matatu as % of motorized commuters | 88% | 95% | 96% | 92% | 96% | 76% |
| Percent adult commuters who walk (a + b + c) | 34% | 35% | 52% | 50% | 45% | 66% |
| Percent adult commuters who walk <10 min (a) | 8% | 7% | 9% | 12% | 8% | 11% |
| Percent adult commuters who walk 10–20 min (b) | 16% | 23% | 25% | 15% | 25% | 37% |
| Percent adult commuters who walk ≥20 min (c) | 9% | 6% | 17% | 21% | 12% | 16% |
| Percent adult commuters who spend ≥30 min each way | 54% | 48% | 31% | 43% | 36% | 26% |

Overall, 60% of adults are employed. Men are significantly more likely to be employed than women. Male employment rates by city range from 73% to a high of 85%, while women's employment rates range from 31% to a high of 48%. Mombasa is an outlier, reporting both the lowest female employment rate (31%) and the highest male employment rate (85%).

Among working-age adults, 71% commute for work or study. In all cities outside of Nairobi, Mombasa, and Eldoret, the majority of commuters walk. Strikingly, among these walk commuters, most walk less than 20 min (Figure 4). Across all of urban Kenya, we find that 30% of all commuters walk less than 20 min, and 11% walk for 20 min or more. This finding stands in contrast to previous studies that have reported that urban Africans walk for hours each day either because there are few motorized alternatives available or because they cannot afford them.

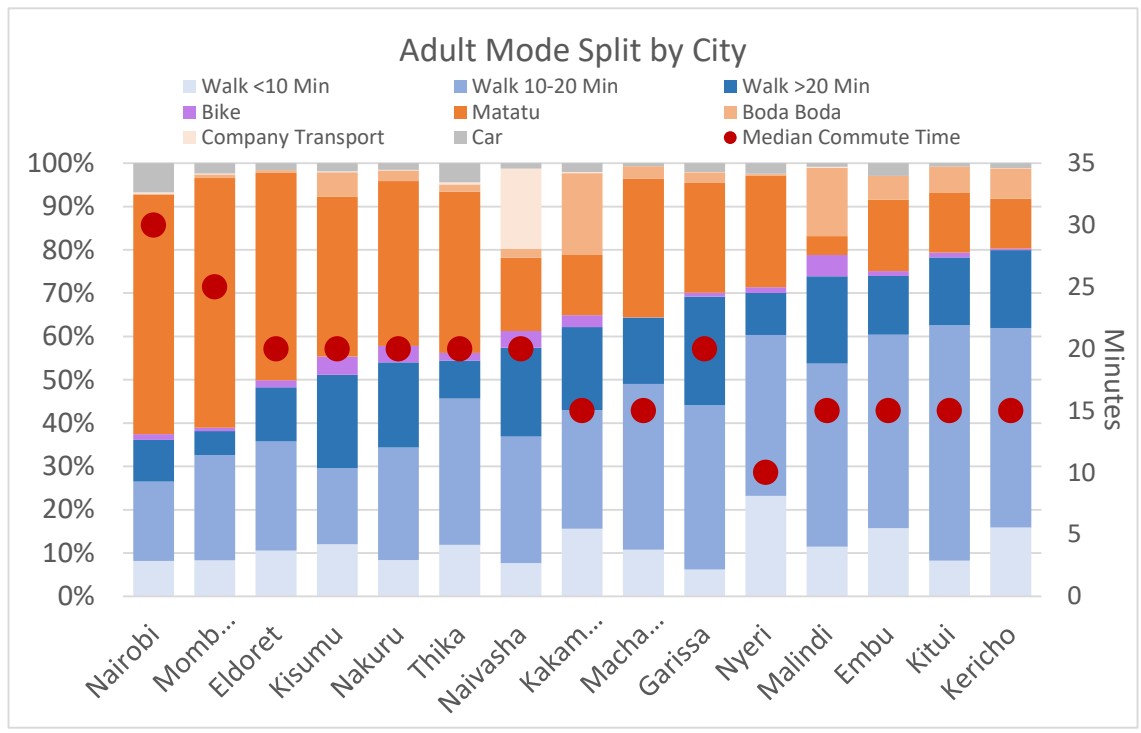

**Figure 4.** Adult mode split and median commute time by city.

The use of motorized transport is widespread in the biggest five cities, but low in the smaller cities. As indicated earlier, in Nairobi and Mombasa, the majority (60–62%) rely on motorized modes for commuting. In Kisumu, Nakuru, and Eldoret, the share of motorized commutes ranges from 37–48%. In smaller cities, the share of motorized trips drops—only about a quarter of all commutes are on motorized modes. It is worth noting that this analysis focuses only on commute travel. The 2010 ACET survey data show that walking is even more prevalent when looking at trips for all purposes; 72% of all trips recorded in that survey were walk trips [29,33].

Figure 4 illustrates the variation in commute mode split between Kenyan cities for adults. It reinforces the dominance of walking in most cities, with Nairobi and Mombasa being the only cities where fewer than 40% of adult commuters walk. In both cities, long walk commutes are relatively uncommon.

Matatus are, by far, the most used form of motorized transport—they account for almost 88% of all motorized commutes in the 5 biggest cities, and 72% of the motorized trips in the 10 medium-sized and small cities. Figure 4 also shows the relative share of other modes of transport. Use of personal cars is highly limited, with Nairobi (7%) and Thika (5%) recording the highest usage. In Naivasha, 19% of commuters report a reliance on company transport. The use of boda bodas (shared motorcycle

and bicycle taxis) is significant in six smaller towns, especially in Kakamega and Malindi (20% and 16%, respectively).

Figure 4 also depicts the median adult commute time in each city. There is a clear trend that median commute times decrease as the percent of commuters walking increases. In other words, commuters using motorized modes spend longer on their commutes than do walk commuters. We discuss this finding and its implications in more detail in Section 7.

Next, we examine whether use of motorized transport is systematically different between the following two groups: (a) those residing in formal vs. informal settlements, and (b) men vs. women. Table 5 summarizes the bivariate relationships between motorized commuting and these variables. Residents of informal settlements commute using motorized modes less often than residents of formal settlements, and this difference is statistically significant in 11 out of 15 cities sampled.

**Table 5.** Fraction of adult commuters using motorized modes by category and city.

|  | Formal | Informal | Male | Female |
|---|---|---|---|---|
| Eldoret | 54% * | 44% * | 52% | 49% |
| Embu | 21% ** | 10% ** | 20% | 18% |
| Garissa | 28% ** | 75% ** | 28% | 30% |
| Kakamega | 16% ** | 5% ** | 14% | 17% |
| Kericho | 14% ** | 7% ** | 15% ** | 10% ** |
| Kisumu | 44% | 37% | 43% * | 35% * |
| Kitui | 16% | 10% | 17% | 14% |
| Machakos | 31% | 34% | 32% | 31% |
| Malindi | 7% | 3% | 8% ** | 3% ** |
| Mombasa | 66% ** | 55% ** | 66% ** | 56% ** |
| Nairobi | 68% ** | 50% ** | 63% | 59% |
| Naivasha | 36% * | 44% * | 36% | 38% |
| Nakuru | 42% ** | 26% ** | 39% | 44% |
| Nyeri | 29% ** | 8% ** | 29% | 27% |
| Thika | 49% ** | 8% ** | 45% ** | 38% ** |

Note: Asterisks and shading indicate a statistically significant difference between categories: 0.05 level = ** and darker shading, 0.10 level = * and lighter shading.

One finding especially worthy of note is that in 10 of the 15 cities there is no substantial difference between men and women in their use of motorized transport. In 5 cities, women are systematically less likely to commute using motorized transport. A 2010 analysis of Nairobi slum resident travel found that even after controlling for factors such as childcare responsibilities and education level, women were significantly less likely than men to commute using motorized transport [26]. While we still see a relationship between gender and employment, these new data suggest that gender may not be clearly related to the mode split among those who do commute; we explore this relationship further using multivariate analysis in Section 6.

Figure 5 illustrates the relationship between educational attainment and the mode split in the five largest Kenyan cities and for the Rest of Urban Kenya. It reveals a strikingly strong relationship—walking drops and motorized transport use rises steadily with increasing education levels, and this holds across each of the five cities and the RUK. Household income is correlated with motorized commuting as well, but the relationship is weaker; this relationship is not shown in a figure.

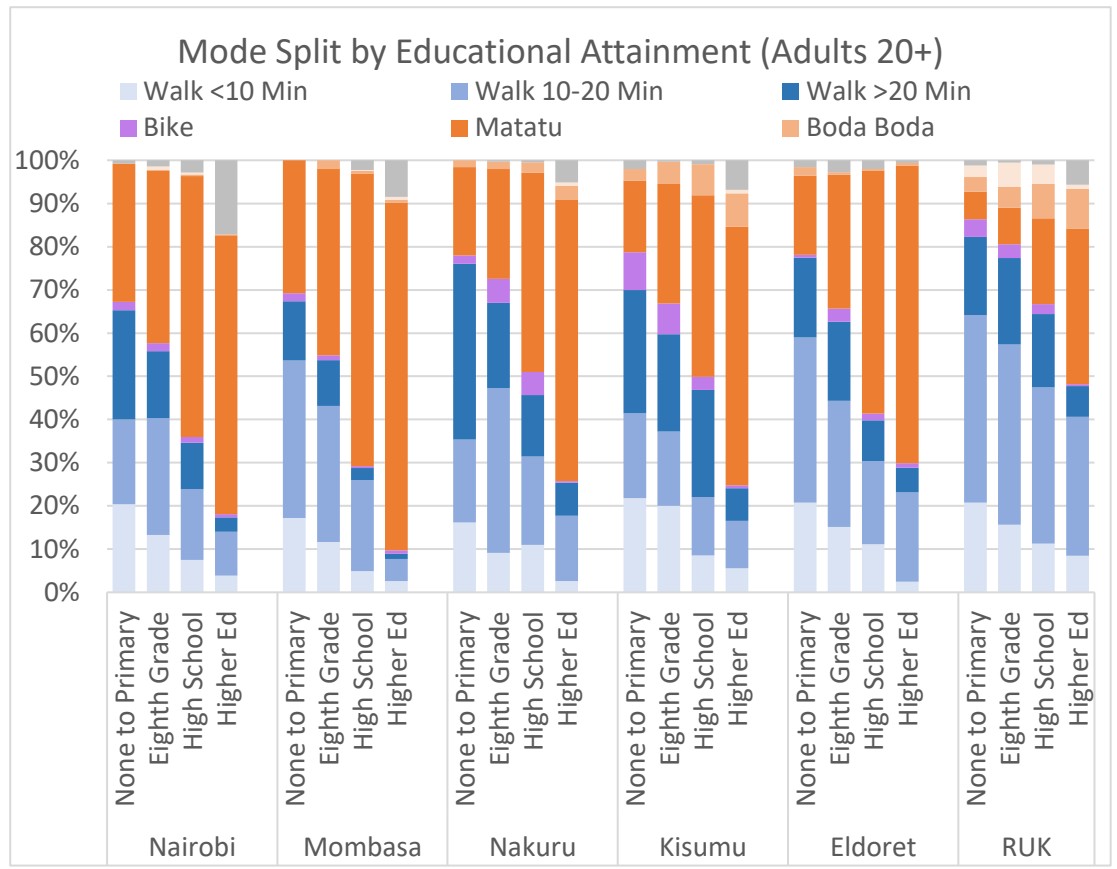

**Figure 5.** Adult mode split for five largest Kenyan cities and the rest of urban Kenya—comparison across educational attainment levels.

## 6. Multivariate Results: Commute Status and Commute Mode

Bivariate relationships provide some strong hints as to which factors are the major determinants of whether a person commutes in urban Kenya and the mode that they use. Multivariate statistical models can reveal more, however. In particular, multivariate models allow us to estimate relationships between outcomes (in this case, commuter status and commute mode) and individual covariates while controlling for all other covariates included in the model. Multivariate models also quantify the estimated strength of these relationships, valuable for potential policy application of the results.

Here, we present the results of multinomial logit models of the joint "choice" of commuter status (whether a person commutes) and commute mode (how they commute) for survey respondents between 18 and 60 years of age in urban Kenya. Multinomial logit models allow us to conduct statistical analyses that predict categorical dependent variables. They work by predicting the probability that each observation will be in each category based on a set of explanatory variables. The specific functional form of the choice probabilities is as follows [34]:

$$P_{ni} = \frac{e^{\beta' x_{ni}}}{\sum_J e^{\beta' x_{nj}}} \tag{1}$$

where

- $P_{ni}$ is the probability of individual $n$ choosing alternative $i$,
- $\beta$ is a vector of estimated coefficients,
- $x_{ni}$ is a vector of explanatory variables for each individual $n$ and alternative $i$,
- and $j$ indexes the alternatives in the choice set.

In these models, the adults in each estimation sample fall into one of the following three categories: (a) Non-commuter; (b) walk commuter; and (c) matatu commuter. Because car commuters are so few in number in Kenya, but their characteristics differ substantially from all other commuters, they are excluded from this analysis. Including commuter status as one of the categories allows us to see that some explanatory variables are associated with whether people commute at all, while others are associated with the modes that commuters use.

Models are estimated separately for Nairobi, Mombasa, three mid-sized Kenyan cities, and for the remainder of the urban Kenya sample, and the unit of analysis is the individual adult. The explanatory variables include household-level information such as slum residency and household income, presence of small children (aged 0–4), presence of schoolchildren (aged 5–14), and the home's (Euclidean) distance from the city center. Also included are several individual-level demographic characteristics such as gender, age category, and level of educational attainment. Observations are weighted in the analyses using weights that are specific to the Kenyan census enumeration area where each household lives. Stata 14 software was used for all analyses in this article, including these models.

Table 6 presents the average estimated effects of switching discrete explanatory variables' values from "0" to "1" on the probability of each outcome, and, for household income, the effect of adding 10,000 Ksh in income per month. Some of the effects are estimated separately for each gender (denoted with "F" and "M" in Table 6), to account for the fact that certain socioeconomic factors have different effects for men and women. These were obtained using the *margins* command in Stata, which provides a weighted average of the individual effects of changes in the explanatory variables. Those effects that are estimated separately for women and men are averaged only across the relevant subsample in the data.

Interpretation of estimated discrete effects is straightforward and informative. For instance, the Eldoret, Kisumu, and Nakuru model estimates that, all else equal, being male (as opposed to being female) means that the probability of being a non-commuter drops by 31 percentage points, the probability of being a walk commuter increases by 16 percentage points, and the probability of matatu commuting increases by 15 percentage points. Note that because the probabilities of being in each category must always sum to 1, the changes in these probabilities must sum to 0 (e.g., 15 + 16 − 31 = 0). In many cases in Table 6, however, the effects do not perfectly sum to 0 because small effects are not reported due to lack of statistical significance.

Where statistically significant, the direction of the estimated relationships confirms expectations. Higher incomes, education, age, and distance to the city center are associated with more matatu use. In contrast to the findings of the bivariate analysis in Section 5, this multivariate analysis shows that after controlling for associations with other explanatory variables, residence in an informal settlement has a negative relationship with matatu use only in Nairobi. The presence of small children in the household has a strong positive relationship with non-commuter status for women, but this relationship is weaker and in the opposite direction for men. This is expected, since women are often responsible for the care of small children, and men in their households are probably trying to cover for extra child-related expenses.

Comparing the magnitudes of the estimated effect sizes, these models illustrate that educational attainment has an especially large positive relationship with matatu commuting. Interestingly, the dominant negative relationship differs between men and women. For women, higher education is most strongly negatively associated with non-commuter status. For men, however, higher education is most strongly negatively associated with walk commuting. This reflects the reality that men of all education levels are likely to be commuters, but higher education is associated with matatu use. On the other hand, women with little education are likely to simply stay at home.

**Table 6.** Effect of socioeconomic factors on adult commuter status and commute mode.

| | Nairobi | | | Mombasa | | | Eldoret, Kisumu, Nakuru | | | Smaller Cities in Kenya | | |
| --- | --- | --- | --- | --- | --- | --- | --- | --- | --- | --- | --- | --- |
| | N = 1681 | | | N = 1751 | | | N = 4199 | | | N = 12,063 | | |
| | No Comm. | Walk | Matatu | No Comm. | Walk | Matatu | No Comm. | Walk | Matatu | No Comm. | Walk | Matatu |
| Informal Settlement | | | −0.05 ** | −0.06 * | 0.04 * | | | | | −0.02 * | 0.03 ** | |
| Income (10,000 Ksh) | | −0.03 *** | 0.03 *** | 0.03 ** | −0.09 *** | 0.06 *** | | −0.04 *** | 0.04 *** | | −0.02 *** | 0.02 *** |
| Male | −0.29 *** | 0.09 *** | 0.20 *** | −0.51 *** | 0.15 *** | 0.35 *** | −0.31 *** | 0.16 *** | 0.15 *** | −0.38 *** | 0.30 *** | 0.08 *** |
| Child <5 in HH (F) | 0.20 *** | | −0.15 *** | 0.14 *** | | −0.11 *** | 0.17 *** | −0.08 *** | −0.09 *** | 0.12 *** | −0.10 *** | −0.02 ** |
| Child <5 in HH (M) | | −0.08 ** | 0.11 *** | −0.05** | | | | | | −0.07 *** | | 0.06 *** |
| Child 5–14 in HH | | | | | | −0.05 * | | | | | −0.04 *** | 0.02 *** |
| <8th Grade (base) | - | - | - | - | - | - | - | - | - | - | - | - |
| 8th Grade (F) | | | | | | | −0.08 ** | | 0.08 *** | −0.10 *** | 0.09 *** | |
| High School (F) | −0.14 ** | | 0.12 * | | 0.06 * | | −0.12 *** | −0.07 * | 0.20 *** | −0.17 *** | 0.10 *** | 0.07 *** |
| Higher Ed (F) | −0.34 *** | | 0.35 *** | −0.49 *** | | 0.42 *** | −0.28 *** | | 0.35 *** | −0.41 *** | 0.22 *** | 0.19 *** |
| 8th Grade (M) | | | | | −0.22 *** | 0.28 *** | −0.10 *** | | 0.13 *** | −0.06 *** | | 0.05 *** |
| High School (M) | −0.14 ** | −0.15 * | 0.29 *** | | −0.41 *** | 0.43 *** | −0.08 ** | −0.20 *** | 0.28 *** | −0.04 * | −0.09 *** | 0.13 *** |
| Higher Ed (M) | −0.17 *** | −0.22 *** | 0.39 *** | | −0.54 *** | 0.60 *** | −0.09 ** | −0.31 *** | 0.40 *** | −0.08 *** | −0.18 *** | 0.25 *** |
| Age 18–24 (base) | - | - | - | - | - | - | - | - | - | - | - | - |
| Age 25–34 (F) | −0.12 *** | 0.10 *** | | | | | −0.11 *** | 0.07 ** | 0.05 * | −0.11 *** | 0.13 *** | −0.02 * |
| Age 35–44 (F) | −0.14 ** | 0.21 *** | | | 0.08 * | | −0.19 *** | 0.12 *** | 0.06 * | −0.14 *** | 0.16 *** | −0.03 * |
| Age 45–60 (F) | | | | | | | −0.12 ** | | 0.13 *** | −0.07 ** | 0.10 *** | −0.03 * |
| Age 25–34 (M) | −0.13 *** | | 0.12 *** | −0.25 *** | | 0.19 *** | −0.16 *** | | 0.11 *** | −0.14 *** | 0.11 *** | 0.03 ** |
| Age 35–44 (M) | −0.14 *** | | 0.16 *** | −0.28 *** | 0.10 ** | 0.18 ** | −0.16 *** | | 0.11 *** | −0.14 *** | 0.10 *** | 0.04 ** |
| Age 45–60 (M) | −0.14 *** | | | −0.19 *** | | 0.20 ** | −0.13 *** | | 0.11 ** | | | |
| Dist <2 km (base) | - | - | - | - | - | - | - | - | - | - | - | - |
| Distance 2–4 km | | | 0.15 ** | | | 0.13 ** | | −0.06 * | 0.05 * | | −0.07 *** | 0.05 *** |
| Distance 4–8 km | | −0.23 *** | 0.29 *** | | | 0.13 ** | 0.06 * | −0.14 *** | 0.08 *** | | −0.07 *** | 0.05 *** |
| Distance >8 km | | −0.24 *** | 0.26 *** | | | | 0.09 * | −0.10 * | | | −0.07 ** | |

Notes: Statistical significance indicated using asterisks: *** $p < 0.01$, ** $p < 0.05$, * $p < 0.1$. Empty cells indicate statistically insignificant marginal effects. Multicity models also include city indicator control variables. Adults were excluded from this analysis if the data were missing key variables, most importantly household income.

A striking point that emerges from this analysis is that Mombasa is somewhat different from other Kenyan cities with regard to commuting differences between genders. Women are much less likely than men to be commuters at all, and only the highest levels of educational attainment affect the probability of commuting for women in this city. In addition, educational attainment has an especially large effect on commute mode for men in Mombasa; even an 8th grade education makes a Mombasa man 28 percentage points more likely to be a matatu commuter, compared to someone with fewer years of schooling.

Distance from the city center has a positive association with matatu use and a negative association with walk commuting, but little relationship with whether an adult commutes at all. This contrasts somewhat with findings reported by Nakamura and Avner [35] that job access declines dramatically with distance from the city center in Nairobi unless car access is available.

The sustainability implications of these results are important. As incomes levels rise and cities spread geographically, we should expect to see moderate-to-large increases in the matatu commute mode share and commensurate decreases in the walk commute mode share. Higher education will also increase the demand for matatu commuting, but in this case, these models predict that much of the increase will be due to an increase in employment—especially for women. These effects are much smaller in magnitude in Kenya's smaller cities, where even walk commutes are short.

As matatu demand rises in Kenya's large cities, increasing pressure will be put on an already stressed road network, with associated reductions in travel speeds and increases in commuting time. Combating this trend and maintaining the sustainability of Kenya's urban transport systems will require strategies to both enhance the pedestrian experience and provide improved public transport service.

## 7. Commute Times are Shorter than Expected, and Walkers Have Shorter Commutes than Motorized Commuters

The final piece of the commuting picture is how long people are spending on their commutes. Looking at all commuting, we see that the average commute time in urban Kenya is on par with what we see in many US cities—aggregating across the 15 Kenyan cities, the reported commute time is 27 min, on average. Only in Nairobi is the mean adult commute time greater than 30 min one way. These findings are consistent with matatu traffic model predictions for a 2015 Nairobi scenario reported by Gonzales et al. [23].

Table 7 provides average commute times by mode for the five largest Kenyan cities and the RUK. Notice that matatu trip times are substantially longer than walk commute times in urban Kenya—in some cases by a factor of two. This is consistent with findings from the 2013 JICA survey for Nairobi, which reported average walk times of 32 min together with average car and matatu commute times of 52 and 58 min, respectively. While the JICA commute time estimates are longer than we find using the World Bank 2013 survey, the ratio between modes is approximately the same; walk commuters in Nairobi spend only about half as much time commuting as motorized commuters. Even if we acknowledge that congestion on city streets slows down motorized transport, the significantly shorter commute time for walkers indicates that they are traveling shorter distances. Unfortunately, this survey did not include questions about commute distance, destination, or time of day; these are questions for future research.

Figure 6 focuses attention on the adult commuters in each city who report commuting more than 30 min each way to work. The green portions of the bars indicate the fraction of commuters that report commuting 30 min (light green) or less (dark green) each way to work by any transport mode. The remainder of the bars divide those commuters who report commuting more than 30 min each way into the modes that they use.

**Table 7.** Adult worker commute time by mode and city.

| City | Commute Time (Adults) | | Average Commute Time by Mode | | |
|---|---|---|---|---|---|
| | Average | Median | Walk | Matatu | Car |
| Nairobi | 31 min | 30 min | 21 min | 38 min | 36 min |
| Mombasa | 26 min | 25 min | 16 min | 32 min | 30 min |
| Eldoret | 24 min | 20 min | 22 min | 26 min | 17 min * |
| Kisumu | 24 min | 20 min | 21 min | 29 min | 25 min * |
| Nakuru | 23 min | 20 min | 21 min | 25 min | 19 min * |
| RUK | 20 min | 15 min | 18 min | 29 min | 24 min |

* These averages for car users are based on sample sizes of fewer than 20 commuters.

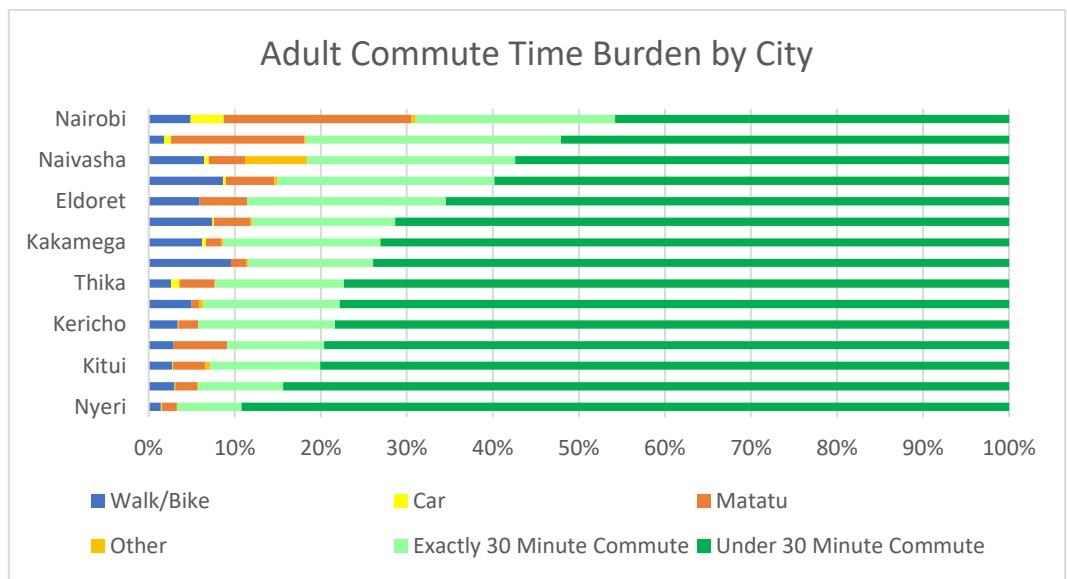

**Figure 6.** Adult commute time burden by city.

The first point that emerges from Figure 6 is that in most Kenyan cities, it is rare to commute more than 30 min each way to work or school. Nairobi's commuters are the most likely to be in this category, but it still accounts for only one-third of Nairobi commuters. In most of the smaller cities included in this survey, fewer than 10% of commuters spend more than 30 min each way on their commutes. Note also that a large fraction of survey respondents reported commuting exactly 30 min each way to work; there is almost certainly a substantial amount of rounding that is happening here.

This extent of time-burdened commuters is similar to what is found in a number of cities in the United States. In fact, Nairobi commute times reported in this survey are actually somewhat shorter than commute times in both New York City and Los Angeles in the United States—especially comparing travel times for public transport (matatus in Nairobi) [36]. The key difference, however, is that American commuters are largely traveling in relative comfort by car, whereas most Nairobi commuters are either walking or riding a matatu.

A second point is that the percent of walk commuters who are time-burdened is much lower than that of motorized commuters in most Kenyan cities. Clearly, walk commuters appear to be traveling only short distances for work or school. This is encouraging, as it suggests that most urban Kenyans have organized their lives such that they do not need to walk-commute long distances. By way of comparison, Olvera, Plat, and Pochet also report that although a large fraction of trips in six other African cities were made on foot, only 10–20% of those walk trips were 30 min or longer [37]. These results were based on surveys from the 1990's and early 2000's that were conducted in somewhat smaller cities in West Africa.

In Nairobi, the fact that 12% of walk commuters regularly walk more than 30 min each way (and an additional 13% report walking exactly 30 min each way) is cause for concern given that the streets are not safe for pedestrians and lack basic infrastructure such as sidewalks and crosswalks. Many prior studies of transport in Nairobi have called for increased investment in pedestrian infrastructure, and these results strongly support this suggestion.

Finally, it appears that in a few of the smaller cities, a large fraction of matatu commuters do have high time burdens. In these cities, recall that the fraction of total commuters who use matatus is small; the cities are small places, so people accessing local jobs generally walk. Many of those who do use motorized transport appear to be traveling quite long distances—likely to destinations outside of their home cities—to access jobs. Therefore, although the time burden may appear similar for matatu commuters in small and large Kenyan cities, the challenges faced by these commuters are quite different. Nairobi matatu commuters are mostly accessing jobs in that city, but it takes them a long time because Nairobi is physically large and the roads are heavily congested. Matatu commuters in small cities have long trips in large part because many are travelling long distances, reporting 2–3 h commute times.

## 8. Children's Commute to School in Urban Kenya

Thus far, this paper has focused entirely on adult commuting in urban Kenya. The survey collected information on school commuting choices of 9974 children as well, which is worth reporting on briefly. The story is simple: Everywhere in Kenya, schoolchildren under 18 walk for most of their commutes, and most report 10–20 minute walk times. There is some variation; only 74% of Nairobi children walk, compared to 95% of the children in Kitui. Those who do not walk almost universally report traveling by matatu. As with adults, children riding to school in a matatu report longer travel times than those who walk.

## 9. Estimating the Size of the Matatu Market

Matatus are widely used, but have been heavily criticized for their poor record on safety, affordability, and emissions. They also, however, provide relatively low-cost motorized transport without government subsidy. In response to a need for better options, there is a lot of activity in the public transport space in Nairobi. The government is planning a roll-out of bus rapid transit [38], and companies such as Uber, Little, and Swvl are beginning to offer app-based luxury bus commuting services [39,40]. Meanwhile, the organized matatu industry is fighting the competition [40]. To provide context and to appreciate the complexity of the challenge, understanding the size of this market is critical.

How big is the matatu market in terms of trip volume and value? To estimate the size of the market, we include commutes for work and study both by working-age adults (18–60 years of age) and school-age children (5–18 years of age). Table 8 reports our estimates. Extrapolating from matatu use in the sample to the population of each city, we estimate a total of 1.53 million commuters using matatus daily for work or study. Commuters in Nairobi and Mombasa account for most of these—almost one million and quarter million, respectively. The smallest cities report 5000–7000 matatu commuters, with Malindi reporting as few as 2200 matatu commuters.

To calculate annual matatu fare revenues in each city, we assume that each matatu commuter makes one round trip each day, 5 times a week, 39 weeks in a year for schoolchildren, and 50 weeks in a year for adults. We then calculate the market size using both median and mean fares for each city.

**Table 8.** Size of the matatu market: trips and fare revenues, by city.

| | Percent Matatu Commute | Median Matatu Fare (Ksh) | Mean Matatu Fare (Ksh) | Population | Daily Matatu Commuters (No.) | Matatu Median Daily Fare Revenue Estimate (Ksh) | Annual Matatu Market w/Median Fare ($US) | Annual Matatu Market w/Mean Fare ($US) |
|---|---|---|---|---|---|---|---|---|
| | (a) | (b) | (c) | (d) | (e) = (a × d) | (f)=2 × (a × b × d) | (g) | (h) |
| Nairobi | 31% | 50 | 50 | 3,133,518 | 956,126 | 95,612,550 | 273,427,210 | 275,296,359 |
| Mombasa | 27% | 40 | 41 | 938,131 | 251,631 | 20,130,460 | 58,373,524 | 60,417,181 |
| Kisumu | 18% | 30 | 51 | 388,311 | 69,782 | 4,186,938 | 11,764,150 | 20,033,446 |
| Eldoret | 26% | 20 | 40 | 289,380 | 75,344 | 3,013,740 | 8,568,603 | 17,260,808 |
| Nakuru | 18% | 30 | 36 | 307,990 | 56,916 | 3,414,948 | 9,831,137 | 11,705,148 |
| Machakos | 17% | 30 | 67 | 150,041 | 25,774 | 1,546,450 | 4,267,729 | 9,487,247 |
| Thika | 19% | 25 | 44 | 136,917 | 26,276 | 1,313,792 | 3,760,497 | 6,663,811 |
| Naivasha | 8% | 30 | 47 | 169,142 | 13,761 | 825,674 | 2,349,226 | 3,651,628 |
| Nyeri | 13% | 30 | 39 | 119,353 | 15,915 | 954,902 | 2,660,074 | 3,428,597 |
| Kitui | 5% | 110 | 85 | 109,568 | 5,660 | 1,245,208 | 3,485,217 | 2,679,222 |
| Kakamega | 7% | 50 | 68 | 91,768 | 6,736 | 673,614 | 1,848,345 | 2,519,139 |
| Garissa | 10% | 30 | 35 | 116,317 | 11,373 | 682,384 | 1,894,562 | 2,232,312 |
| Embu | 11% | 50 | 56 | 60,673 | 6,605 | 660,528 | 1,817,123 | 2,025,326 |
| Kericho | 6% | 50 | 54 | 101,808 | 6,226 | 622,560 | 1,765,963 | 1,900,265 |
| Malindi | 2% | 60 | 77 | 118,265 | 2,246 | 269,538 | 755,777 | 970,969 |
| Total | | | | | 1,530,370 | | 386,569,137 | 420,271,458 |

Notes: (1) Mean fare calculation includes only reported fares under Ksh 300. (2) Population is from Kenyan census 2009 and includes Core Urban + Peri Urban Population). (3) Exchange rate Ksh 84.5/USD using the average for 2012.

We find that the annual matatu market in these 15 cities, taken together, is roughly about $387–420 million in revenue, depending on whether we use median or mean fares. Nairobi is, by far, the largest matatu market with annual commute trips generating an estimated $273–275 million in fare revenues. These estimates are conservative, given that they are: (a) Based on 2009 population, while Kenya's population increased at an estimated 2.7% annually from 2009–2013 [41]; and (b) exclude all other kinds of trips such as those for leisure, shopping, and medical care.

## 10. Discussion and Conclusions

This paper reports new insights about commuting in urban Kenya based on a large 2013 household survey conducted by the World Bank. Prior data on commuting in Kenya were available only for Nairobi, making these data an important new source of information on other Kenyan cities. Analyzing a relatively large sample of 26,204 of working-age adults and 9974 school-age children, we investigated who is commuting and who is not, the modes commuters use, and how much time and money they spend. We focus on understanding the implications of our results for urban transport sustainability—now and in the future—in both Nairobi and in smaller Kenyan cities.

### 10.1. What Determines Whether Urban Kenyans Commute, and Which Modes They Use?

Walk commuting is prevalent throughout urban Kenya. In mid-sized Kenyan cities, approximately half of adult commuters walk, and in over half the cities surveyed, more than 70% of adult commuters walk. Nairobi and Mombasa—Kenya's two largest cities—are the only cities where walk commuters comprise less than 40% of the adult commuting population. The vast majority of children commuting to school also walk.

Over 90% of commuters in urban Kenya report availability of motorized public transport (matatus), but use and household-level "effective access" is lower. Specifically, 60% of commuters in Nairobi and Mombasa, 40%–50% in Kenya's mid-sized cities, and only 25% of commuters in the smaller cities use a matatu for their commute.

To better understand who commutes and how, we developed four separate multinomial logit models for Nairobi, Mombasa, the three mid-sized cities (Nakuru, Kisumu, and Eldoret), and RUK. For working-age adults (18–60 years), we find that the likelihood of commuting at all is systematically lower for women and, for both genders, it increases with education and age. Presence of small children in the household strongly reduces the likelihood that women will commute, while increasing the likelihood that men will commute.

Given that the vast majority of commuters essentially choose only between walking and riding a matatu, what factors help explain their mode selection? The logit models show that matatu use increases with education and age, and also increases with income. Distance to the city center also matters—the farther one lives from the city center, the more likely he or she is to use a matatu. These results suggest that the gap between availability and actual use of matatus is partly a reflection of the fact that some commuters simply do not travel far and can walk, and that for some commuters matatus are unaffordable. The extent to which this gap is a reflection of inadequate service or lack of destination coverage is a question for future research.

### 10.2. How Much Time Are Urban Kenyans Spending on Their Commutes?

The most surprising findings from these data pertain to commute times. Commuters in the smaller Kenyan cities have relatively short commutes; the median commute time is just 20 min. In Nairobi, the median commute time is 30 min, and only 5% of adult commuters in Nairobi reported commuting longer than an hour each way. This finding is a significant departure from the prevailing narrative about excessively long commute times in Kenya's largest city. In every city surveyed, motorized commute times were substantially longer than walk commutes; a relatively small minority of commuters walk more than 20 min each way, but the average duration of motorized commutes was much longer.

Reflecting on what these commute time data mean for urban planning practice in Kenya, we note that there are two ways to solve transportation problems. One is to move commuters more quickly and safely through the city. The other is to reduce the need for urban travel. Given the undisputed severity of Nairobi's traffic congestion, the 2013 survey evidence suggests that most Nairobi residents are solving their transportation problems by limiting their commutes to short distances, off-peak hours, or both. An important area for further research would be to gain a better understanding of how much households are limiting their choice of where to live and work in order to avoid an arduous commute, and what that means for housing affordability and quality as well as access to jobs.

*10.3. What Are the Policy Implications of This Study?*

Kenya's urban commute mode split is enviable from an environmental sustainability point of view. The current public transport system is also financially sustainable—private matatu operators provide a motorized service that is widely available, extensively used, and profitable without any government subsidies. In fact, our estimates show that matatu operators have annual revenues in the range of US$ 386–420 million; this helps explain why they have organized to form a strong lobby to resist government efforts to either influence the matatu market or invest in other forms of public transport.

The problem with the current transport system is that both walk commuters and those riding matatus face several challenges. Walk commuters have relatively short commute times, but are often subject to dangerous travel conditions during their journeys. Matatu commuters, meanwhile, endure substantially longer travel times. In addition, both the literature and media point to problems such as a poor safety record, high emissions, vehicles that are neither comfortable nor well-maintained, and the overloading of passengers.

An important question, then, is whether and how services for and the experience of urban commuters can be improved, while maintaining this sustainable mode split. This is especially important given that these cities are developing and, with rising incomes and education levels, residents' demand for motorized options will also rise. Indeed, Kenya's National Bureau of Statistics data indicate that both private car and motorcycle registrations are increasing rapidly in the country [42]. One promising direction would be to invest in infrastructure for pedestrians, such as sidewalks and footpaths, lighting, and safe crosswalks [3,43,44]. These features are notably absent in much of urban Kenya today, presenting a clear opportunity for significant improvement.

Another important area would be to plan for and invest in public transport in all cities with the explicit goal of maintaining its role as the most widely used mode of motorized transport. This will entail investment in expansion to keep pace with demand volume, service improvements that enhance safety, comfort, and reliability, and the development of mechanisms that can enhance affordability for poor residents. While the government can opt to invest in developing new public transport systems, such as a new bus system or a bus-rapid-transit system, it should consider working with the matatu industry on a reciprocal basis. That is, government could facilitate expansion of matatu systems and services in various cities, in exchange for improvements in safety and certain service indicators. For example, the government could consider incentives that facilitate the switch to cleaner fuels/vehicles, safer vehicles, and transparent subsidies for operating on less profitable routes. The matatu industry could agree to adopt a better system for route planning and coordination, test innovations such as integrated fare cards or vehicle tracking systems that can provide real-time alerts to users regarding arrival of the "next matatu," and agree to certain safety and emission norms.

Broadly, the County governments and national agencies influencing development in Kenyan cities should aim to achieve the following four transport goals: maintain the highly sustainable modal split, improve safety for all road users, transition to cleaner vehicles to improve urban air quality, and improve transport affordability for the poor. The path each city will follow will be unique, but these goals are universal. Achieving them will put Kenya on a path to improving the productivity and quality of urban life.

**Author Contributions:** The authors confirm contribution to the paper as follows: conceptualization, D.S. and S.G.; methodology, D.S. and S.G.; formal analysis, D.S.; funding acquisition, S.G.; writing—original draft preparation, D.S.; writing—review and editing, D.S. and S.G.

**Funding:** This research has benefited from funding from the Global Partnership for Output-Based Aid (GPOBA) and the World Bank. All opinions are those of the authors and not of the sponsoring agencies.

**Conflicts of Interest:** The authors declare no conflict of interest.

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
