# Peer review of "Commuting in Urban Kenya: Unpacking Travel Demand in Large and Small Kenyan Cities"

_sustainability, doi:10.3390/su11143823_

Round 1
Reviewer 1 Report
Dear Authors,
Comments are provied in the attached file.

Reviewer 2 Report
This is a high quality submission and a potentially significant contribution to the field. However my main reservation is that it currently reads as a 'study' rather than an academic journal paper framed by specific theoretical issues. I would therefore advise that the authors pay particular attention to the opening sections, to articulate the research problem. Although the literature on transport in Kenya / East Africa is limited, I do feel that there are relevant articles on Sub-Saharan Africa and elsewhere across the Global South that could help provide a more robust grounding, and therefore more insightful analysis than the baseline study that the paper currently represents. In addition, I would also recommend the following:
Methodology - provide a bit more detail and critique about how the census surveys were conducted, and the questions asked.
Estimating the size of the Matatu Market - although this is interesting, it seems to fit a little oddly because there is no foregrounding earlier in the paper.
Time lapse - it has been quite a long time since the original study was conducted. Although some of the more recent literature has been cited, I feel that it would be useful for the authors to provide some further commentary on relevant developments since 2013.
Reviewer 3 Report
Authors of the paper present results of data analysis that were collected under contract to the World Bank in 2013 and the results of multinomial logit models of the joint “choice” of commuter status (whether a person commutes) and commute mode for survey respondents between 18 and 60 years of age in urban Kenya. The subject of the paper is interesting and worth investigating. The methodology of the research is rather understandable, but it raises some doubts about the quality of the results and some issues require further clarification.
1. In the opinion of the authors, is the research sample sufficient? (what percentage of the population in each category of household or individuals included?)
2. What could be the reason for the significant differences in the studies presented (Table 1) and which studies are reliable and representative?
3. Do the authors mean both private motorized modes and matatus by motorized mode? It would be useful to clearly define the terms "motorized mode" and "motorized trip", if they do not appear in the paper in different meanings. It should be clearly defined whether the authors in the various parts of the paper refer to private motorized or public motorized transport (public transport is usually motorized).
4. What's the meaning of the phrase “Gap between availability and use of public transport” (table 3)?
5. Please comment on the value of Pseudo-R2 (table 6)
6. Please develop information on multinomial logit models by providing examples of calculation equations.
7. What could be the cause of congestion in cities (only 1-8% of households own a car)? Are matatus causing congestion?
Round 2
Reviewer 1 Report
Dear Authors,
Thank you for providing response for my comments. I'm sure that this paper will be an important contribution within its research area.
Author Response
Thanks for your words of support! We have made small additional edits to the draft in response to one reviewer's comments.
Reviewer 2 Report
The authors have addressed the previous reviewer comments commendably. My final request would be to restate and address the research questions that you set out on p2 directly in the conclusion (this should be simple and quick to do - you have all of the material there anyway).
Author Response
Thanks for your words of support! We have made small additional edits to the conclusion in response to your comments.
Reviewer 3 Report
There are several messages “Error! Reference source not found” in the delivered pdf.
Author Response
Apologies for the "Error" messages. They have been cleared up. We have also made small additional edits to the draft in response to one reviewer's comments.